# KABEDONN: POSTHOC EXPLAINABLE ARTIFICIAL INTELLIGENCE WITH DATA ORDERED NEURAL NETWORK

## ABSTRACT

Different approaches to eXplainable Artificial Intelligence (XAI) have been explored including (1) the systematic study of the effect of individual training data sample on the final model (2) posthoc attribution methods that assign importance values to the components of each data sample. Combining concepts from both approaches, we introduce kaBEDONN, a system of ordered dataset coupled with a posthoc and model-agnostic method for querying *relevant* training data samples. These *relevant* data are intended as the explanations for model predictions that are both user-friendly and easily adjustable by developers. Explanations can thus be finetuned and damage control can be performed with ease.

## 1 INTRODUCTION

Although machine learning (ML) algorithms are not expected to be perfect, their unexplained failures can be detrimental e.g. the well-known incident of 'racist' algorithm bbc (2015). EXplainable Artificial Intelligence (XAI) has emerged as an effort to help improve trust in the use of ML algorithms. It is a burgeoning field that has been recently studied from different aspects, such as (1) data influence on model training (2) post-hoc attribution methods (3) "signal methods" etc (some methods fall into multiple categories as seen in surveys like Arrieta et al. (2020); Gilpin et al. (2018); Tjoa & Guan (2020); Adadi & Berrada (2018)). With improved trust, powerful blackbox models like the deep neural network (DNN) can be adopted into real applications with more accountability. Combining some of these existing concepts, we introduce k-width and Bifold Embedded Data Ordered Neural Network (kaBEDONN), which is a post-hoc XAI method to query *relevant data* as the explanation for a model prediction. All python codes are available in the supp. materials.

Here, we consider the image classification task, including experiments on common image datasets MNIST, CIFAR10, ImageNet. Denote a sample data as $(x, y0) \in D = X \times Y$ where $X$ is the input space and $y0 \in Y$ the ground-truth class label. $x$ is classified using some base model $f$ as $c = argmax_i(y_i)$ where $y = f(x) \in \mathbb{R}^C$ and $C$ the number of classes/categories. The scenario considered in this paper is the following: users wish to know why $f$ labels the sample as $c$ i.e. they require explanations for the predictions. Like many XAI methods, kaBEDONN aims to provide a form of explanation. We start by clarifying our three objectives.

**Objective 1. Relevant data as explanations**. The relevance of data has been measured in different ways. In Koh & Liang (2017), a training data sample is considered either *helpful* or *harmful* to the prediction made by a trained model, quantified by the *influence* score. In Yeh et al. (2018), data samples are either *excitatory* or *inhibitory*, in Pruthi et al. (2020) *proponent* or *opponent*. For kaBEDONN, relevant data samples strongly activate main nodes or sub-nodes, hence, they are excitatory in a different sense than Yeh et al. (2018). Here, explanatory images are considered relevant when their features look "similar" to $x$ according to the base model $f$. The explanatory images are then presented to users as shown in fig. 1(A) and fig. 2(A).

More technically, we have three different contexts of "similar". (1) A *representative data* $r$ is a training data sample that has been used to construct a main node in kaBEDONN. In this case, kaBEDONN stores the processed signals of $r$ (also called "fingerprint" in (Tjoa & Cuntai, 2021)) and $r$'s index w.r.t the ordered dataset in a *main node*. (2) A *similar data* $s$ is a training data sample that has neither been included as a main node nor a *sub-node* because it is already well-represented

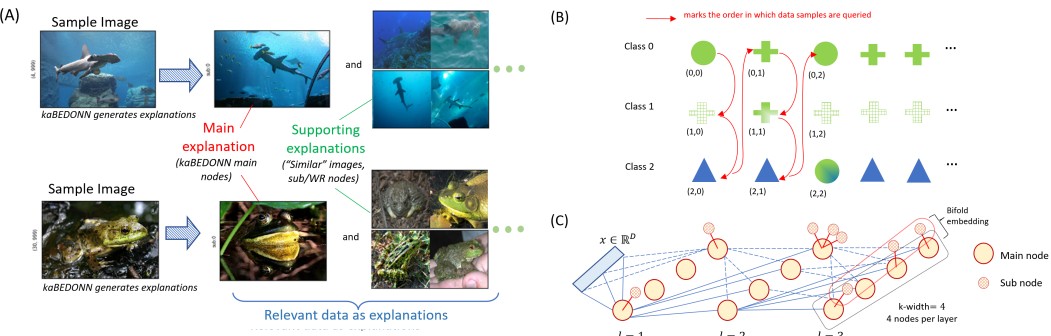

Figure 1: Inspired by *influential* images for explainable AI Koh & Liang (2017); Yeh et al. (2018); Pruthi et al. (2020), kaBEDONN provides explanations by querying *relevant* images from ordered data. (A) Sample images from the ImageNet dataset and the *relevant data* queried by kaBEDONN as explanations. (B) Data samples ordered by class $y0$ and position index $idx$. Data are queried in a deterministic order during kaBEDONN construction. (C) kaBEDONN with 3 layers and $kwidth = 4$. Bifold embedding: the first fold consists of layers of main nodes, the second "upper" fold consists of sub-nodes. WR nodes are not shown.

by an existing main node. Only the index of $s$ (but not its fingerprint) is stored in a *well-represented (WR) node* (that belongs to some main node). (3) A *boundary data* is a training data sample that is similar enough to a representative data $r$ but is different (they have different class labels); this happens, for example, when similar-looking breeds of cats are labeled differently. Its fingerprint and index are stored in the sub-node of the main node constructed from $r$.

**Objective 2. Adjustment of Explanation for debugging**. Explanations do not necessarily convince every user to the same degree. Suppose users flag some explanations as unsuitable. kaBEDONN is a system that allows developers to quickly finetune explanations based on the given feedback, primarily by adding, removing or reordering the underlying data samples used for kaBEDONN construction. This is useful because problematic data samples (e.g. accidentally mislabeled data and overly-representative data*) can be identified and removed while better explanations can be incorporated to the system when available. *Remark*. *An example of overly-representative data is an optical illusion; since it appears different from different point of views, it is not helpful as an explanation.

*A real user feedback example from ImageNet*. In fig. 2(B) panel 1, the image of interest (bullfrog image) activates a main node with the wrong class label (hammerhead shark). Furthermore, we found that a cartoon image of a hammerhead shark in the ImageNet dataset is associated with that node. Suppose a user considers it undesirable, we need the developer to quickly readjust the explanation. This is done by simply reordering the cartoon image (panel 2): we push it to the back of the queue by renaming the image. kaBEDONN is then reconstructed (panel 3), and a more "similar" representative is presented (the node appears to respond to partially dark background). We have deliberately chosen the unclear and ambiguous bullfrog image from ImageNet to demonstrate how problematic case can be handled. The result is thus not perfect; in practice, iterative user/developer feedback may be needed for better result. Also see appendix *General info* for more remarks.

**Objective 3. Predictive correctness flag for debugging**. kaBEDONN is a posthoc explanatory model complementary to a more complex blackbox base model $f$. It is constructed using the collection $\{(x_e, y0) : x_e = f_{enc}(x), (x, y0) \in D' \subseteq D\}$ where $x_e$ is a latent vector obtained from encoder $f_{enc}$. The encoder can be $f_{enc} = f$ itself or only the latent encoder part of $f$. In this paper, $x_e = y = CNN(x)$ i.e. $f_{enc} = f$ for the simplicity of demonstration. kaBEDONN can also act as a predictive model through multi-layered processing of latent vectors, partially using the universal approximation (UA) concept in Tjoa & Cuntai (2021). While kaBEDONN no longer has the UA property from Tjoa & Cuntai (2021), the data fitting capability is still very high (see experiment and result section). The discrepancy between predictions made by kaBEDONN and the base model $f$ (e.g ResNet) serves as a flag for user/developer to report abnormal prediction.

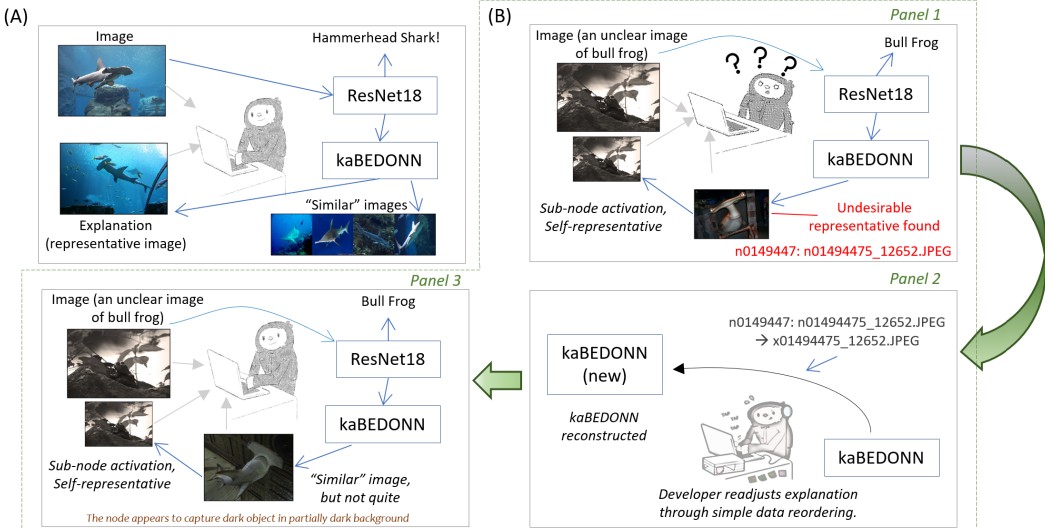

Figure 2: kaBEDONN provides explanations for ImageNet samples. Gray arrows are what users see. (A) kaBEDONN provides a user with a representative image as the standard explanation and "similar" images as supporting explanations. (B) kaBEDONN applied on *an ambiguous image of a bullfrog*. User finds an explanation (through sub-node activation) with undesirable main representative: cartoon image of hammerhead shark in ImageNet dataset. Developer fixes the issue by reordering the undesirable data. New representative is queried. It is still not quite the right explanation although it has similar colour texture (dark object in dark background). Furthermore, with kaBEDONN's sub-node activation, self-representative sub-node is activated, thus yielding the correct label.

To summarize, kaBEDONN is not only a posthoc XAI method that presents users with "similar" images, but is also an adjustable system that is friendly to model developers for two reasons. (1) *Adjustment of $D'$*. Problematic samples can be easily readjusted, reducing the likelihood that they turn into representative kaBEDONN main nodes (i.e. less likely served as explanations), thus improving the quality of main explanations. Explicit indexing $(y0, idx) = (class, index)$ of the data makes adjustment/feedback process easy. (2) kaBEDONN is posthoc and model agnostic, hence it does not directly interfere with the main model $f$. By comparison, Neural Backed Decision Tree Wan et al. (2021) requires finetuning of the main model's weights. Regardless, kaBEDONN helps developers collect user feedback on some sample data; data flagged as problematic may later be excluded from further finetuning of $f$.

## 2    RELATED WORKS

**Post-hoc attribution methods**. XAI methods such as Local Interpretable Model Agnostic Explanation (LIME) Ribeiro et al. (2016), SHapley Additive exPlanation (SHAP) Lundberg & Lee (2017), Class Activation Mappings (CAM) Zhou et al. (2016), Layerwise Relevance Propagation (LRP) Bach et al. (2015) and their variants compute the attribution values of the components of individual data. These values represent the components' "importance" to the final prediction. For image data, the components are pixels. These methods are posthoc: all computations are made *after* model training/finetuning, thus they do not change any of the model's components. Our model is also posthoc, but instead of computing the importance of data components, we compute the importance of the data w.r.t the adjustable, ordered dataset underlying kaBEDONN.

**Data influence**. Koh & Liang (2017); Yeh et al. (2018); Pruthi et al. (2020) compute the *influence* of training data on the final model. The most influential training sample is then shown as the explanation to the prediction of a sample. kaBEDONN similarly presents *relevant images* as explanations; our corresponding influence score being the strength of neuron activation. The influence score in Yeh et al. (2018) is called the *representer value*, obtained through representer theorem optimization

Schölkopf et al. (2001). A data point is then considered inhibitory/excitatory if it suppresses/augments the activation value. kaBEDONN is also in a sense excitatory: if $x$ is represented well by $r$, then $x$ will strongly activate a node that has been constructed using $r$. More precisely, our influence score is measured as the strength of node activation measured in $[0, 1]$: strong activation is near 1, and weak activation is near 0 i.e. it has an excitatory aspect; kaBEDONN does not explicitly define inhibitory response. Furthermore, training sample $r$ that has been converted into kaBEDONN node is influential in the sense that it is able to represent a number of other samples $WR(r)$ (defined more precisely later).

In extreme cases, the influence of a training data is computed by comparing the model that has been trained with a given data and another model that has been trained without the same given data Koh & Liang (2017). This is a prohibitively expensive process, thus Koh & Liang (2017) modified the influence functions (a method from statistics) for efficiency. The effect of *upweighting* a sample on the resulting model parameter is computed, from which its *influence* is obtained. Pruthi et al. (2020) introduces TracIn, inspired by Hara et al. (2019), which also measures the "influence" of training samples on a particular test sample by computing the difference in loss between predicted and actual test sample before and after training. TracIn is thus non strictly posthoc, in the sense that the model $f$ is altered after TracIn training. By contrast, kaBEDONN is posthoc since $f$ is left intact, and can be finetuned independently based on feedback gathered from kaBEDONN.

**SQANN**. Our model kaBEDONN is a direct modification of Semi-Quantized Activation Neural Network (SQANN) Tjoa & Cuntai (2021). SQANN has been developed as a universal approximation (UA) of functions in which each node represents the $k$-th training data $(x^{(k)}, y^{(k)})$ in an ordered dataset. Activating the node associated to sample $k$ strongly (i.e. $[v_l^{(k)}]_i \to 1$) causes SQANN to fetch the $\alpha$ value that corresponds to $y^{(k)}$, yielding the correct output. SQANN is constructed by testing one sample after another from an ordered dataset. kaBEDONN similarly draws samples in a sequential manner, although our system specifically organizes dataset by classes; see fig. 1(B).

Like SQANN, neuron activation in kaBEDONN is also computed with *double selective activation* $\sigma_{dsa}$. Given selective activation $\pi(x, a) = \frac{a}{a+x^2}$ and Super Gaussian $s_g(x, a) = exp(-(x/a)^{2n})$, $n = 4, r = 0.5$, then $\sigma_{dsa}(x, a_1, a_2, r) = (1 - r) \times \pi(x, a_1) + r \times s_g(x, a_2)$. This is used by Node object (see appendix, pseudo-code 2).

*Nodes and $\alpha$ values*. Similar to SQANN, kaBEDONN stores fingerprints of samples in nodes and stores their corresponding labels as $\alpha$ values. When strong activation is detected (i.e. $act > \tau_{ad}$), the $\alpha$ value of the strongly activated node is fetched for the computation of output. Interpolations are also performed for weak activation, and the interpolation method is customizable. In this paper, the interpolation choice is simple: $\alpha$ value of the node with the strongest activation will be used (even when its activation does not exceed $\tau_{ad}$).

*Differences from SQANN*. Although SQANN is constructed to remember training data output perfectly, it comes with some caveats. For example, layer size is not controlled. A layer can grow arbitrarily large and imbalanced. kaBEDONN, on the other hand, is designed with $k$-width, i.e. each layer has $k$ nodes (except possibly the last layer) as the name suggests. Furthermore, it has a "bifold embedding" structure: kaBEDONN has layers of main nodes/neurons like DNN, but each node may contain one or more sub-nodes. Collectively, the sub-nodes form the second fold of layers as shown in fig. 1(C). The UA property is no longer available in our model, replaced with the capability to identify explanatory training data samples in the three contexts previously mentioned.

## 3 KABEDONN

In this section, we introduce our main model and illustrate it with the donut toy data. kaBEDONN's parameters are adjustable but, here, we focus on presenting the concept and only test a few particular choices like $a_1, a_2, r = 0.1, 0.5, 0.2, \tau_{ad} = 0.5, \tau_{act} = 0.95$ (admission and activation thresholds respectively). Optimization of parameter choices are left for future studies. This section is descriptive; more details are available in pseudo-codes and full codes in the main code repo.

At the core of kaBEDONN, some technical specifications are briefly described as the following (see the appendix for more details). A *node* object consists of (1) the *main node* and its key (*main key*): they are the fingerprint and index $(y0_{node}, idx_{node})$ of a data sample respectively (2) the well-

represented nodes $WR_{li}$ and (3) the sub-nodes $S_{li}$ where $l$ denotes layer and $i$ the i-th node in that layer. Other concepts include the *normalized node-relative vector* that store information as a relative vector rather than full vector, elastic set of size *kwidth* (to limit layer sizes) and finetuning. In essence, they serve to distinguish nodes that are representative of certain classes from sub-nodes that correspond to more ambiguous data. kaBEDONN helps to identify such data for the users.

**Ordered dataset for classifications**. Dataset is ordered in folders, where each folder corresponds to a class label $c$ (interchangeably $y0$). This is compatible with ImageFolder arrangement from pytorch. For a class $c$, the perceived order of data samples depends on python $sorted$ function. For a given set of parameters and data ordering, the construction will be deterministic. We denote a data sample with its index $(y0, idx) = (c, idx)$ or with its input $x^{(y0, idx)}$, whichever convenient in the context. As illustrated in fig. 1(B), data samples are put into a queue in a "transverse" manner: fetch the first samples of all classes, then the second samples of all classes and so on (unless the class has no more samples). This ordering helps kaBEDONN to query explanatory data precisely and quickly, making the explanation adjustment process easier.

A **LayerNodes** object is an ordered collection of nodes in layer $l$ of kaBEDONN that has an assemble_receptors method. A *receptor* $R_l$ is a matrix assembled from the fingerprints of all main nodes in the layer. It is used in the intermediate computation of the activation $v_L$ of some layer $L$; see appendix pseudo-code 3. More precisely, each $R_l$ is used in the $l$-th iteration of kaBEDONN forward propagation, shown below in its recursive form:

$$f_{kab}(x, l) := v_l(x) = \sigma_{dsa}(||(v_{l-1}(x) - R_l)/N_{l-1}||)$$ (1)

where $x$ is the input, $N_l = |v_l| + \epsilon$ and $\epsilon$ a small number to prevent division by zero (i.e. for smoothness). The term $||(v_{l-1} - R_l)/N_{l-1}||$ is called *normalized stimulation*.

**kaBEDONN Training**. The training is essentially a series of LayerNodes constructions. In each layer, the goal is to build an elastic set with size $kwidth$ i.e. to choose main nodes, as the following. As seen from filter_ by_ layer_ activation() function in pseudo-code 1, a training sample $(y0, idx)$ is admitted into the elastic set if it does not activate any previous nodes that have been chosen as the candidate main nodes i.e. all $i$ components of activation $[v_l]_i < \tau_{ad}$. Let us call this process the *admission filtering*, illustrated in appendix fig. 6(A).

*kaBEDONN's bifold embedding*. Recall that SQANN Tjoa & Cuntai (2021) achieves UA by an extreme form of restructuring: all layers beyond the collision layer $l_c$ are destroyed, and the sample $x_c$ that causes collision is then added to $l_c$ so that it contains a fingerprint that will overwhelm the activation that results in collision and inaccurate prediction. kaBEDONN is not as extreme. A sample not used as main node is tested for bifold activation. This mechanism either (case 1) sets the sample $x_c$ aside as a well represented node if its label matches the main node's, see appendix fig. 6(B), dotted-arrow numbered 6 or (case 2) embeds the nrvec of $x_c$ as a sub node (appendix fig. 6(B), dotted-arrow numbered 5,9); this is similar to collision in Tjoa & Cuntai (2021), and usually occurs to aforementioned *boundary data*. Future stimulation by $x_c$ will strongly activates this sub-node, causing it to return the $y0$ (corresponding to $x_c$) that has been stored as $\alpha$, hence yielding the correct prediction.

**Obtaining representative data samples as explanations**. Finally, how do we obtain explanations like fig. 1(A)? In essence, forward propagation (see pseudo-code 1) is used to retrieve the explanations. When strong activation in the main node or sub-node are detected, the fingerprints stored in respective nodes are used to fetch the relevant data. Otherwise, interpolation is performed (in this paper, the most strongly activated node is used to fetch the main and supporting explanations).

**Donut Toy Data for Illustration**. We apply kaBEDONN with $kwidth = 8$ on 120 data points. Each data point $(x, y0) \in \mathbb{R}^2 \times \{0, 1, 2\}$ resides within a dataset with donut-shaped distribution as shown in fig. 3(B); see full version in appendix fig. 8(A). Each point also belongs to one of the three classes 0,1,2, coloured black, dark grey and light-grey respectively. Used as a predictive model, the accuracy performance of this particular kaBEDONN instance is $1.0/0.925$ on training/test datasets respectively, although larger kaBEDONN easily fits the data perfectly. The final model has 3 layers (appendix table 2). Out of 120 data points, 20 are automatically filtered and embedded into the model. Also see appendix subsection on Toy Donut Data for more details.

How exactly are main nodes, *well-represented* nodes and sub-nodes distributed? See fig. 3(A,B). A main node is labeled as a magenta triangle and it represents several other data samples in its prox-

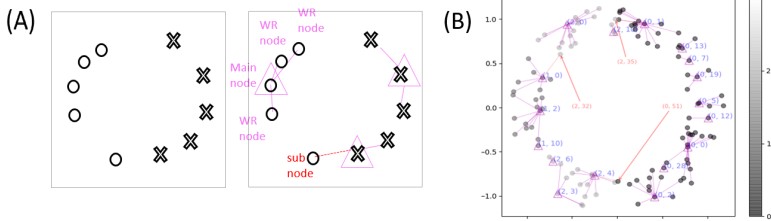

Figure 3: (A) Illustrative mini toy donut. (B) Donut toy dataset with three labels (0 black, 1, grey, 2 light grey). Best viewed electronically (zoom in; also, full version in appendix).

Table 1: Accuracies comparison when kaBEDONN is applied as a predictive model with various *k* widths. CNN denotes the original performance (no kaBEDONN).

|  | kwidth | 16 | 32 | 48 | 64 | CNN |
|---|---|---|---|---|---|---|
| mnist | train acc | 0.987 | 0.996 | 0.999 | 0.999 | 0.948 |
|  | test acc | 0.888 | 0.913 | 0.925 | 0.926 | 0.951 |
| cifar | train acc | 0.965 | 0.993 | 0.998 | 0.999 | 0.861 |
|  | test acc | 0.681 | 0.722 | 0.721 | 0.722 | 0.776 |

imity. All data samples linked to this particular main node by magenta lines are *well-represented*, hence they are called well-represented (WR) nodes. As the name suggests, a training sample used by kaBEDONN as a main node is expected to represent its locality well i.e. it has the same label as its WR nodes.

*Sub-nodes*. However, "locality" is not always well delineated. See for example fig. 3(A) "sub node". The sample belongs to the circle class but stays near an x i.e. it is the so-called *boundary data*, close enough to activate the x's main node. kaBEDONN bifold embedding is exactly used to resolve such situations: the circle at the boundary might be embedded into a sub-node in the "upper fold". See appendix for more explanations on the full donut dataset.

*Explanations for XAI*. kaBEDONN quickly tells users which training data are "similar" to the data point being tested. These representative samples act as the explanations. Although there is no individual image like fig. 1(A) to be shown for toy donut data, kaBEDONN points to the directory of the explanatory data (text version of explanations printed by kaBEDONN like in appendix fig. 10). See appendix for more details on (1) fig. 3(B) where we explain more kaBEDONN nodes w.r.t the entire dataset with visual references (2) main and supporting explanations on donut data similar to fig. 1(A).

## 4 EXPERIMENTS AND RESULTS

We apply kaBEDONN on MNIST, CIFAR and ImageNet datasets. In fig. 4 of Koh & Liang (2017), 900 images from two ImageNet classes (dog and fish) are used to compute their influences on test samples. We refer to this process as the **class-selective explanation**. We perform similar but modified experiments.

**Convolutional Neural Network (CNN) for MNIST**. MNIST is a relatively simple dataset thus we use a simple CNN with 3 successive conv+ReLU modules followed by conv+AveragePool+FC (fully connected layer) with 4 epochs of training, batch size 16 (details in the available code). The training/test accuracies attained are 0.948/0.951 respectively. kaBEDONN is constructed using $\{(x_e, y0) : x_e = CNN(x) : (x, y0) \in D\}$.

**MNIST with class-selective explanations**. For MNIST, we demonstrate kaBEDONN with class-selective explanation on class 4 and 9 here, especially because handwritten 4 and 9 might look similar. We can thus see the effect of node and sub-nodes activations clearly. Fig. 4 shows samples with different types of kaBEDONN neuron activations. First MNIST sample (first row) shows a

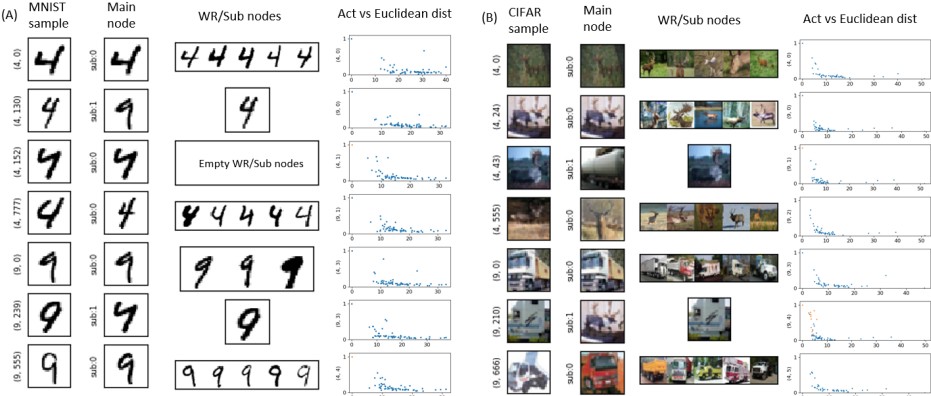

Figure 4: Display of image samples associated to main/WR/sub nodes for XAI. Best viewed electronically (zoom in) (A) MNIST (B) CIFAR. sub:1(0) indicates that a subnode is (not) activated. If sub:0(1), then WR(sub)-nodes associated to the main node are displayed. Act vs Euclidean dist is similar to fig. 4 of Koh & Liang (2017), except each plot is for a single sample and activations of neurons, *act*, are our "influence" values. Orange dots are for the activations of sub-nodes. It shows only a loose trend: shorter Euclidean dist. appears to be weakly correlated to higher activation.

standard activation: the very 1st training sample activates the main node that corresponds exactly to itself. The 2nd sample is a digit 4 that activates the main node of a digit 9. However, thanks to bifold activation mechanism, the sample correctly activates the sub-node that refers to itself (hence marked sub:1), yielding a correct prediction. The 4-th sample activates a main node corresponding to another digit 4. While they do not necessarily look the same visually, they are similar enough according to the CNN. The 6-th sample is similar to 2nd sample, and 7-th to 4-th. Act vs Euclidean distance plots like Koh & Liang (2017) are also shown, although there is no conclusive trend to be found.

**Relevant data samples for XAI**. As previously mentioned, we present *relevant data* as the explanations for a model's prediction. In fig. 4, whenever no sub-node activation occurs (sub:0), the image associated with the main node is presented as the explanation, and the associated images in its WR nodes are the "supporting" explanations. Otherwise (sub:1), the main node is only "similar but not quite the same", and the sub-node is presented as the actual representative explanation.

**kaBEDONN with the whole MNIST dataset**. Used as a predictive model, the accuracies of kaBEDONN models with different $kwdith$ is shown in table 1. Compared to its base model CNN, kaBEDONN overfits the training dataset. Fortunately, *they are able to maintain reasonably high test accuracies* given certain hyperparameters, i.e. kaBEDONN has generalization power. This is a very promising prospect as it opens up a new research direction: is there a latent encoder different from CNN that can be used with kaBEDONN to improve both training and test accuracies? Although UA property in Tjoa & Cuntai (2021) is not achieved, generally, kaBEDONN's accuracy on the training dataset is high and improves with $kwidth$.

**CIFAR**. More details on similar experiments on CIFAR can be found in the appendix A.4. Some XAI results are also shown in fig. 4. As a predictive model, the accuracy results are also equally promising (table 1). Furthermore, a brief look at fig. 4 CIFAR shows a promising trend for further research: for example, row 1 and 2 both classify images of a deer, but the first main node clearly reflects a deer in a green background with plants/grass. With a good latent encoder, kaBEDONN demonstrates a good potential for fetching meaningfully similar images as explanations.

**Data ordering as hyperparameter**. kaBEDONN can be used as a complementary predictive model. For each $kwidth$, different accuracy results (on test data) are obtained when we shuffle the data ordering randomly. The results are shown in fig. 5. This reflects the idea that some data samples are more suitable as main nodes, while some others are more suitable as WR- or sub-nodes. The intuition is clearly illustrated by a boundary data. Suppose a sample data $x_b$ resides near the boundary between different classes. If we accidentally place $x_b$ early in the ordering, it might acci-

dentally become a main node. Many samples around the boundary will activate $x_b$ and this might not be desirable because $x_b$ over-represents different classes. Many redundant sub-nodes may be attached to its main node, although the sub-nodes could have been well-represented by only a few other nodes that provide better representation. However, the suggestion above is only intuitive; we believe there may exist a boundary with extreme variations that may need such over-representation. In any case, fig. 5 implies that it is possible to create a better representation for kaBEDONN through appropriate ordering, and this is a potential direction for further studies.

**ImageNet**. Experiments similar to *class-selective explanations* on ImageNet have been performed. Generally, the results are similar to MNIST and CIFAR (fig. 4), thus we will briefly discuss them here. Fig. 1(A) shows partial results for ImageNet; the full results can be found appendix fig. 11 in which two classes of images are arbitrarily chosen, and 900 images are used each, just like previous similar works in this topic. kaBEDONN works as expected: when there is no sub-node activation (sub:0 scenario), main nodes are queried as explanations along with "similar" images (or the images in the WR nodes). Otherwise (sub:1 scenario), the image used to construct the sub-node is returned. For training data, sub-node activation typically returns the correct class by querying the node associated with the sample itself.

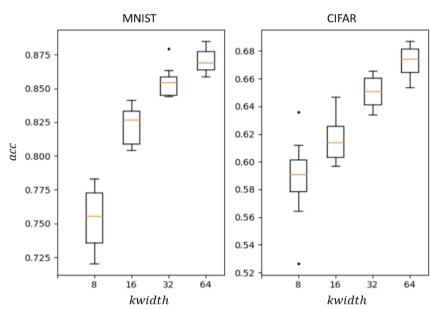

Figure 5: Test Acc. vs $kwdith$. For the same *kwidth*, different accuracies are obtained due to different data ordering.

kaBEDONN model on the entire 1 million images from ImageNet dataset is trickier. Ideally, kaBEDONN is designed to extract a subset of representative images and use them as its representative nodes. However, the large dimensions of features in ImageNet coupled with the large size of dataset have caused some inefficiency. While kaBEDONN has helped resolve the imbalance layer size (with *kwdith* limitation), we believe further improvement in the model can be aimed at improving its filter efficiency. One way is to modify the preprocessing pipeline e.g. by reducing the dimension and subcategorizing classes, which is especially relevant because many classes in ImageNet has overlaps (e.g. many breeds of the same species of animal, dog). A controlled selection of representative data samples could then be performed, e.g. by repeated application of kaBEDONN on ImageNet data subsets and collecting only main node samples. Our temporary results (not shown) indicate that even with smaller dataset, the large number of labels being trained at once into kaBEDONN harms the construction process, further supporting the idea that subcategorizing labels is the better potential solution. Future studies on efficiency will be helpful.

## 5    CONCLUSION

We have introduced kaBEDONN, a posthoc model-agnostic method that queries *relevant data* as explanations for model prediction. Systematic ordering of training dataset is required in our system: we aim to encourage a more transparent and organized way of building a model that puts more emphasis on individual training data. Advantages: (1) explanations are user-friendly since they are queried from familiar dataset (2) explanations are adjustable and easy to finetune e.g. by rearranging data order or removing problematic explanations. Hence, kaBEDONN system improves the query/feedback process in XAI.

We have tested kaBEDONN on raw donut data and images that have been encoded into vectors by CNN. From here, several directions for further studies are available as the following. (1) Systematically study the precise effect of dataset ordering. (2) Modify kaBEDONN such that UA and CF in Tjoa & Cuntai (2021) are recovered. (3) Improve its generalizability so that kaBEDONN not only works as an XAI method, but also as a predictive model. This includes finding the optimal latent encoder (e.g. transformer rather than CNN) that works well with kaBEDONN.

Listing 1: *Pseudo-codes for kaBEDONN*. Some components are in the appendix.

```
1  def fit_data(X,Y):
2      # main kaBEDONN loop
3      # eset: elastic_set
```

```
4    q = create_queue()
5    l_now=1 # current layer
6    eset, altset=[],[]
7    while q is not empty or len(eset)>0:
8      eset,q,info=mould_elastic_set(q)
9      eset,setaside,nodes_layer=\
10       filter_by_layer_activation(eset, l_now)
11     altset.extend(setaside)
12
13     if len(eset)>=k_width \
14       or info['ALL_USED_UP']:
15       integrate_nodes_layer(l_now,
16         nodes_layer)
17       bifold_activation(l_now, altset)
18       eset,altset=[],[] # reset
19       l_now+=1
20
21   def bifold_activation(L,altset):
22     X,Y=fetch_data(altset)
23     for i,(x,y0), in enum(X,Y):
24       anode=get_activated_node(x, L)
25       y0_node,idx=anode.main_key
26       INSERT_SUBNODE if y0!=y0_node
27       if INSERT_SUBNODE:
28         nrvec=compute_nrvec(x)
29         anode.new_subnode(nrvec,y0,idx)
30       else:
31         anode.wr_nodes.append(y0,idx)
32
33   def forward(x):
34     act, act_pre = None, x
35     for l in range(1,1+N_layer):
36       receptors=assemble_receptors()
37       act=normalized_stimulation(act_pre,receptors)
38       if any(act>threshold(l):
39         act_id = argmax(act)
40         # now get activated_node
41         anode=layers[l].node_list[act_id ]
42         y_pred, NODE_INFO = anode.forward(act_pre)
43         OUTPUT_INFO = {act_id, NODE_INFO etc}
44         return y_pred, OUTPUT_INFO
45       update_interpolation_info() # customizable
46       act_pre=act
47     y_pred, OUTPUT_INFO = use_interpolation()
48     return y_pred, OUTPUT_INFO
49
50   def filter_by_layer_activation(eset, l):
51     X,Y = fetch_data(eset)
52   % nodes_layer = LayerNodes()
53     new_eset,setaside=[],[]
54     for i,(x,y0), in enum(X,Y):
55       act=compute_activation(x)
56       INSERT_NODE if len(nodes_layer)==0
57         or if all(act<admission_threshold)
58       if INSERT_NODE:
59         nodes_layer.insert_new_node()
60         new_eset.append(eset[i])
61       else:
62         setaside.append(eset[i])
63     return new_eset, setaside, nodes_layer
```

ACKNOWLEDGMENTS

Anonymous for now

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

# A APPENDIX

## A.1 GENERAL INFO

**Adjustment of Explanation**. Recall that one of kaBEDONN's objective is to allow the *Adjustment of Explanation*. Some relevant points:

1. Through kaBEDONN, users' feedback can now be used to modify the dataset; new dataset can be used to retrain $f$ and the resulting model can be compared for further analysis. In this paper, we present only the core system and algorithm design. We leave contextual user-dependent data analysis for future studies.
2. In fig. 2(B), the bullfrog image appears to activate a node that responds to dark background texture.
3. Unlike the process shown in fig. 2(B), we might want to consider deleting problematic data sample altogether. Furthermore, many more samples can be adjusted before the model $f$ is reconstructed.
4. Notice that a complimentary pipeline to collect users' feedback will be needed; the user interface for displaying explanation is not polished in our code, since we focus mainly on the kaBEDONN model.

**More related work: differentiable Search Index (DSI) Nearest Neighbour**. Data samples in the proximity of each node's locality will activate the node strongly, i.e. kaBEDONN uses the concept of distance. This is reminiscent of the concept of distance in nearest neighbours methods and, like SQANN, it has the properties of instance-based learning approaches. However, nearest neighbours methods have been said to be non-generalizing Pedregosa et al. (2011). Like SQANN, kaBEDONN also has a non-linear multi-layer structure, thus it does not suffer from non-generalizing property.

Tay et al. (2022) has been used to fetch document id (docid) $j$ given a text query $q$. In our system, data index $(y0, idx)$ is our "docid" while kaBEDONN's vector input $x$ is our "text query". In its standard mode, kaBEDONN returns the index of *relevant* data sample $(y0, idx)$ along with the predicted class label.

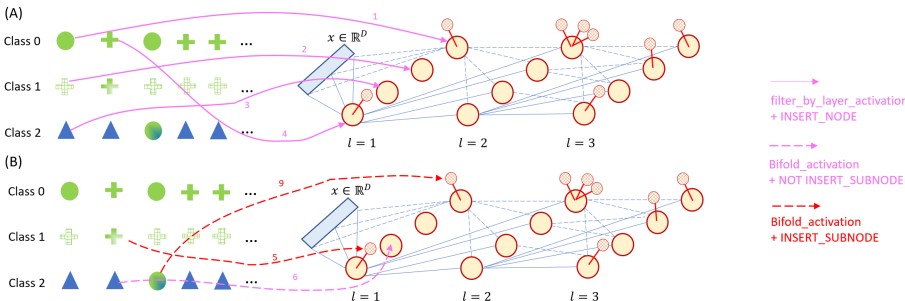

Figure 6: (A) Step 1 to 4. Data samples are admitted into the model through *admission filtering*. The samples are distinct, so they don't activate each other's nodes. (B) Sample 5, 6, 9 are admitted through the mechanisms in bifold activation. Note that sample 5 and 9 are similar to sample 1 and 4 respectively (illustrated with slightly different colours), but they don't share the same classes, hence they are embedded as sub-nodes.

An **elastic set** (denoted as *eset* in pseudo-code 1) is a flexible set in which data indices can be added and removed. The main purpose of elastic set is to provide a running list of data indices that will be used for constructing the *main nodes* of a kaBEDONN layer. Repeated application of filter_by_layer_activation (see pseudo-code 1) is performed until an elastic set of size $kwidth$ is obtained; this helps control the layer size.

A **node** object consists of (1) the *main node* and its key (*main key*): they are the fingerprint and index $(y0_{node}, idx_{node})$ of a data sample respectively (2) the well-represented nodes $WR_{li}$ and (3) the sub-nodes $S_{li}$ where $l$ denotes layer and $i$ the i-th node in that layer. See appendix pseudo-code 2.

As seen from pseudo-code 1, $WR_{li}$ is populated by training samples that trigger bifold activation but require no INSERT_NODE. In other words, each such sample $(y0, idx)$ is well-represented: it activates a main node and labeling the sample with $y0_{node}$ is correct ($y0 = y0_{node}$). The fingerprint of a well-represented sample is not stored in the model, helping the model save memory. As seen in the donut example later, through kaBEDONN's filtering approach, only a fraction of the whole training dataset is eventually embedded into the model.

$S_{li}$ is similar, except $y0 \neq y0_{node}$, in which case a sub-node is required for a more accurate representation of the sample. Intuitively, this occurs at the boundary between classes: the sample activates a main node with a different class $y0_{node}$ (i.e. it is similar enough to activate the particular node) yet they are not the same on closer inspection. The sample's fingerprint is stored in the model. Furthermore, a tunable $\tau_{act}$ will be associated to each sub-node. High activation threshold is used to reduce the likelihood that samples well-represented by the node (labelled $y0_{node}$) accidentally activate this newly admitted sub-node (labelled $y0$) i.e. reduce the likelihood of making the wrong predictions.

Figure 7: Normalized node-relative vector.

A **normalized node-relative vector** (nrvec in pseudo-code 1) is shown in fig. 7. During a sub-node embedding, nrvec is stored as the fingerprint (unlike main node which stores activation $v_l$). Why? Intuitively, two samples that activate the same main node are *similar* i.e. their main vectors are close to each other. Nrvec helps segregate them away to better distinguish their relative positions in the higher dimensional space.

The intuition is simple: we want a layer with main nodes that are representing distinct samples. They might even be from the same class but possess features that are distinct enough from each

other e.g. different breeds of cats. This mechanism thus avoids redundancy in the sense that no two main nodes in the same layers are so similar that they activate each other's nodes.

**Finetuning**. After training, we run through the training data once for finetuning. For each sample, check for the correctness of classification against the ground-truth class, primarily to double-check samples in $WR$ nodes. If it is wrong, finetuning is performed recursively until the prediction is correct. During this process, (1) if a sub-node is activated, then $\tau_{act}$ is increased. The intuition is also simple: we reduces the sensitivity of the sub-node so that wrong samples do not easily activate it (2) if no sub-node is activated, then a sub-node is added into the node (similar to bifold activation with INSERT_SUBNODE). See remark in appendix; also see the main codes.

Fine-tuning in this paper is performed by going through the entire training dataset once, as mentioned in the main text. *Remarks*. (1) the process will terminate because either $\tau_{act}$ will eventually be high enough or a sub-node is added. We exclude unresolvable ill-defined dataset that contains two samples with exactly the same input values but different labels. (2) that UA property in Tjoa & Cuntai (2021) is no longer applicable here i.e. training dataset is no longer perfectly modeled (we empirically show later that the accuracy is still very high). This is because after one fine-tuning run, (A) the increased threshold might cause the node to be too insensitive for samples that previously depend on its activation (B) additional sub-nodes might be too sensitive they are stimulated by previous samples that used to only activate the main node for correct prediction.

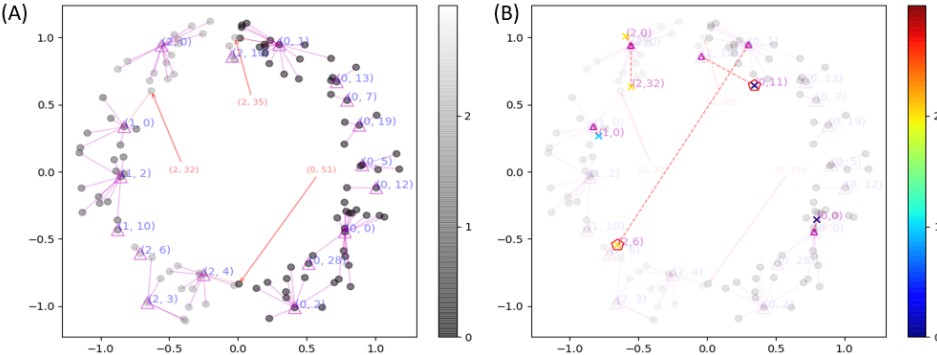

Figure 8: Best viewed electronically. (A) Donut toy dataset with three labels (0 black, 1, grey, 2 light grey). Magenta triangle: a data sample embedded as a main kaBEDONN node. Node linked to a magenta triangle by magenta line is a data sample well-represented by the main node. Node linked to magenta triangle by a dotted red line is a sub-activation node that activated the main node but has a different value. It usually occurs at the class boundary. (B) Test data samples (coloured x marks, magenta $(y0, idx)$) and the main nodes (small magenta triangle) they activated (foreground), superimposed on the transparent version of fig. (A) (background). Red pentagon mark denotes wrongly predicted data.

## A.2 TOY DONUT DATA

Donut data is generated with the following procedure. Draw an array $T$ of size $N$ from a uniform random distribution with $U([0, 2\pi \frac{N-1}{N}])$. Then $X$ is the ordered set containing $X = \{(cos(T_i), sin(T_i)) : i = 1, \ldots, N\}$ and $Y$ is its corresponding label $Y = \{argmax_j(y_j) : i = 1, \ldots, N\}$ where $y = [y_1, y_2, y_3] = [cos(T_i), cos(2T_i), cos(3T_i)]$. The corresponding test data samples are generated similarly by perturbing $X$ with $U([-t_{sd}, t_{sd}])$. In our experiments, $N = 120, t_{sd} = 0.1$.

*Main nodes*. As before, main nodes are magenta triangles in fig. 8. They represent several other nodes nearby.

*Sub-nodes*. As mentioned in the main text, "locality" is not always well delineated. For example, see fig. 8. Node $(0, 51)$ (pointed by a red arrow). The sample has ground-truth label 0, but it is stranded somewhere in the boundary between class label 0 and 2. With existing training samples alone,

```
=== Index:free input! ===
  x: [-1.  -0.2]
  y0: -1
========================
activation:standard
y_pred:1
layer:1
node_idx:5
data_idx:(1, 2)
data_loc:('class1', '17.npy')
parent_node_loc:same as data_loc
```

```
=== Index:free input! ===
  x: [ 0.007 -0.83 ]
  y0: -1
========================
activation:subactivation
y_pred:0
layer:1
node_idx:7
data_idx:(0, 51)
data_loc:('class0', '70.npy')
parent_node_loc:('class2', '110.npy')
```

Figure 10: Sample explanations for donut toy data. Yellow highlight shows filenames of relevant explanatory data samples. This is equivalent to fig. 1(A), but "relevant data", except, rather than images we show the positions of the data in the folder).

there is not enough information to label any samples around the boundary. Furthermore, $(0, 51)$ is close enough to activate main node $(2, 4)$ but it is not well represented, since $(2, 4)$ belongs to class 2. Hence, kaBEDONN's mechanism will embed $(0, 51)$ into the "upper fold": it becomes the sub-node.

*Three types of explanations for XAI* mentioned in the main text. (1) Overarching visual explanation showing kaBEDONN nodes w.r.t the entire dataset can be seen in fig. 8. kaBEDONN quickly tells users which data samples are the representative samples relevant to the data sample being tested. Fig. 8(B) shows test samples (coloured x marks) predicted using kaBEDONN. Small magenta triangles are kaBEDONN *main nodes* activated by the test samples and red pentagons indicate wrong predictions. Each small magenta triangle in fig. 8(B) exactly corresponds to a big magenta triangle in fig. 8(A). For example two yellow x marks i.e. test samples $(2, 0), (2, 32)$ activate a main node the corresponds to $(y0, idx) = (2, 0)$ in the training dataset.

(2) Apart from visual reference, a report is printed to provide users with more explanations, as the following. For example, let us inspect a yellow x marks test sample $(2, 32)$ in fig. 8(B). The explanation for its prediction is reported in the following format $[T/F] sample ==> mainnode$ and a list of "like" nodes e.g. $[T](2, 32) ==> (2, 0)$ and $[(2, 1), (2, 2), (2, 7), \dots]$ (like main and supporting explanations similar to fig. 1(A)). *Like nodes* are simply well-represented nodes corresponding to the main node. For users, they are data points that are "similar" to the user's sample of interest.

(3) In our codes, submode *find* and setting find_ what to *activate_ nodes* allow user to further find the exact folder/name of the training data file that corresponds to the activated node, as shown in fig. 3(C). Yellow highlights show the exact data folder name and file name. During sub-node activation, parent_ node_ loc points to the the exact data folder name and file name for data sample of the related main node.

Figure 9: Similar to fig. 8 except for big donut.

Big donut data is similar, except $y_j = cos(jT_i)$ for $j = 1, 2, 3, 4, 5$ i.e. it has five classes of data samples. Furthermore, $N = 1000$. See fig. 9.

Table 2: Layer information of kaBEDONN constructed on Donut data. $y0, idx$ denotes the indices of data sample that has become a kaBEDONN main node. n(wr), n(sub) denotes the number of well-represented data sample and sub-nodes respectively.

| LAYER 1 | | | | LAYER 2 | | | | LAYER 3 | | | |
|---|---|---|---|---|---|---|---|---|---|---|---|
| y0 | idx | n(wr) | n(sub) | y0 | idx | n(wr) | n(sub) | y0 | idx | n(wr) | n(sub) |
| 0 | 0 | 16 | 0 | 0 | 5 | 4 | 0 | 0 | 28 | 0 | 0 |
| 1 | 0 | 6 | 1 | 2 | 6 | 1 | 0 | | | | |
| 2 | 0 | 14 | 0 | 0 | 7 | 1 | 0 | | | | |
| 0 | 1 | 15 | 1 | 1 | 10 | 0 | 0 | | | | |
| 0 | 2 | 10 | 0 | 2 | 10 | 2 | 0 | | | | |
| 1 | 2 | 11 | 0 | 0 | 12 | 1 | 0 | | | | |
| 2 | 3 | 4 | 0 | 0 | 13 | 3 | 0 | | | | |
| 2 | 4 | 9 | 1 | 0 | 19 | 3 | 0 | | | | |

Table 3: Layer information of kaBEDONN constructed on the whole MNIST data. We show only layer 1, 2 and the last layer, layer 8. Notation is similar to table 2

| LAYER 1 | | | | LAYER 2 | | | | LAYER 8 | | | |
|---|---|---|---|---|---|---|---|---|---|---|---|
| y0 | idx | n(wr) | n(sub) | y0 | idx | n(wr) | n(sub) | y0 | idx | n(wr) | n(sub) |
| 0 | 0 | 989 | 7 | 8 | 1 | 2283 | 429 | 3 | 2102 | 1 | 3 |
| 1 | 0 | 4107 | 4 | 9 | 1 | 2796 | 762 | 2 | 2523 | 0 | 0 |
| 2 | 0 | 106 | 0 | 2 | 2 | 710 | 338 | 4 | 2917 | 2 | 0 |
| 3 | 0 | 844 | 0 | 3 | 2 | 2257 | 8 | 0 | 2949 | 0 | 1 |
| 4 | 0 | 1229 | 0 | 7 | 2 | 529 | 37 | 3 | 3013 | 0 | 3 |
| 5 | 0 | 65 | 12 | 8 | 2 | 277 | 322 | 1 | 3281 | 0 | 1 |
| 6 | 0 | 2953 | 14 | 0 | 3 | 656 | 1026 | 8 | 3617 | 0 | 1 |
| 7 | 0 | 2140 | 0 | 1 | 3 | 1311 | 11 | 9 | 5582 | 0 | 0 |
| 8 | 0 | 338 | 483 | 2 | 3 | 2367 | 152 | | | | |
| 9 | 0 | 78 | 80 | 3 | 3 | 1365 | 5 | | | | |
| 0 | 1 | 298 | 1 | 4 | 3 | 56 | 496 | | | | |
| 2 | 1 | 13 | 0 | 5 | 3 | 925 | 131 | | | | |
| 4 | 1 | 512 | 4 | 7 | 3 | 1094 | 10 | | | | |
| 5 | 1 | 44 | 1 | 8 | 3 | 95 | 199 | | | | |
| 6 | 1 | 757 | 19 | 9 | 3 | 99 | 1175 | | | | |
| 7 | 1 | 652 | 5 | 0 | 4 | 1280 | 14 | | | | |

## A.3   KABEDONN ON MNIST

Most results are already shown in the main text. Using $kwidth = 16$, a kaBEDONN with 8 layers have been constructed. 15147/60000 samples are embedded into the system. Table 3 shows more details on the nodes after kaBEDONN construction.

## A.4   KABEDONN ON CIFAR

For CIFAR, the CNN used is ResNet18 He et al. (2016) with its FC layer replaced to $512 \times 10$ since there are 10 classes in CIFAR. Similar to MNIST, CNN is fine-tuned for 4 epochs on batch size 16 with no further hyperparameter search. Training/test acc. attained are 0.861 and 0.776 respectively.

**CIFAR with class-selective explanations**. CIFAR explanations are reasonably similar to the explanations we see for MNIST, as shown in fig. 4. Sample 1 (row 1) shows a standard activation of main node by a deer image on itself; here, the main node image is served as the explanation. As perviously mentioned, sample 2 is similar, except it shows deer not on a green/plant background. Sample 3 and 6 show sub-activated nodes. The deer/truck activate main nodes of the other class. With

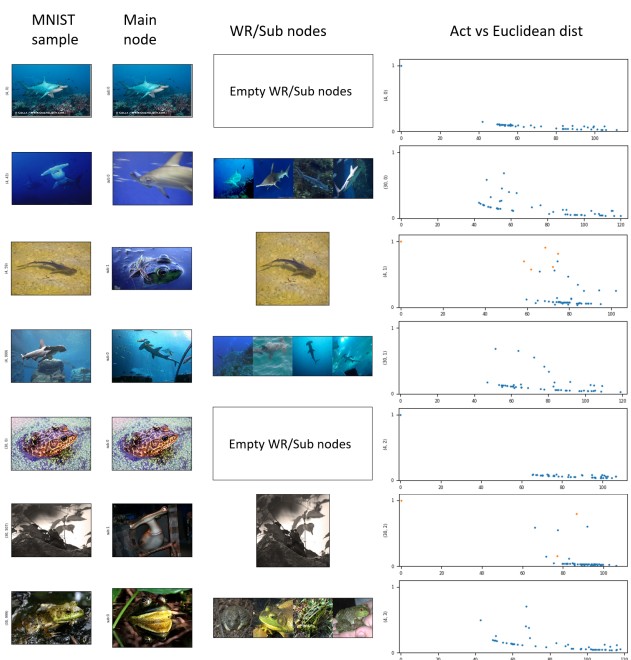

Figure 11: ImageNet display of image samples and kaBEDONN exlpanations, similar to fig. 4

sub-nodes activation, kaBEDONN recognizes the samples as they are. The other samples provide relevant images as explanations as previously explained. Also, Act vs. Euclidean dist plot shows a better pattern; nearer to the sample (i.e. nearer to 0 distance), the activation values tend to be higher.

**kaBEDONN with the whole CIFAR dataset**. The results have also been tabulated in table 1.

## A.5 KABEDONN ON IMAGENET

We perform class selective explanations on two classes in ImageNet, and then show the some results after explanation adjustment.

## A.6 MORE PSEUDO-CODE

Listing 2: Pseudo-codes for kaBEDONN's individual nodes.

```
1  class Node:
2    def init():
3      main_node, main_key = None,None
4      sub_nodes_ = {} # is dict
5      wr_nodes_ = [] # is list
6
7  def assemble_sub_receptors(self):
8    # Node's method
9    # non-memoized version of this function
10   receptors = [] # shape: (n, act_pre dim)
11   temp_idx = []
12   for (y0,idx), nrvec_ in self.sub_nodes_.items():
13     receptors.append(nrvec_['nrvec'])
14     temp_idx.append((y0,idx))
15   receptors = np.array(receptors)
16   return receptors, temp_idx
1  def forward(self, act_pre): # Node's method
2    # act_pre is the activation values
```

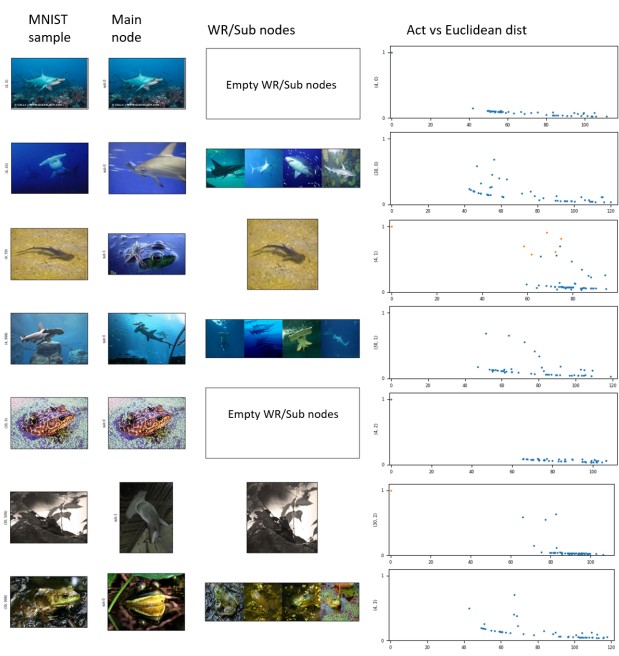

Figure 12: Similar to fig. 11, but after adjustment of explanation is done in fig. 2(B). The results appear to be stable in the sense that the explanations for other images are not changed by a local adjustment.

```
3    # propagated from previous layer
4    y_node,idx= self.main_key
5    nrvec = act_pre - self.main_node
6    nrvec = nrvec/norm(nrvec)
7    # actually, we alsohandle 0 norm
8
9    receptors, temp_idx = self.assemble_sub_receptors()
10   side_act = np.matmul(receptors, nrvec.T)
11   activation_threshold = [subnode['activation_threshold']
12     for _,subnode in self.sub_nodes_.items()]
13   # we do convert it to numpy array
14
15   if np.any(side_act>=activation_threshold):
16     receptor_idx = np.argmax(side_act
17       - activation_threshold )
18     y_pred = temp_idx[receptor_idx][0]
19     NODE_INFO = update_node_info
20   else:
21     y_pred = y_node
22   return y_pred, NODE_INFO
```

Listing 3: Pseudo-codes for kaBEDONN's layer.

```
1  class LayerNodes:
2    def init():
3      node_list = [] # list of Node objects
1  def assemble_receptors(self,):
2    # LayerNodes' method
3    # non-memoized version of this function
4    nodes = []
5    for n in self.node_list:
6      if len(nodes)==0:
7        nodes = [n.main_node]
8      else:
```

```
 9        nodes = np.concatenate(
10          (nodes,[n.main_node]),axis=0)
11    return nodes # convert to numpy array
```

