# OpenReview forum: "kaBEDONN: posthoc eXplainable Artificial Intelligence with Data Ordered Neural Network"
_ICLR.cc/2023/Conference — Submitted to ICLR 2023_

### Official Review · Reviewer_ZXST · 2022-10-22

**Confidence:** 2
**Correctness:** 3
**Technical Novelty And Significance:** 2
**Empirical Novelty And Significance:** 2
**Recommendation:** 3

**Clarity, Quality, Novelty And Reproducibility:**

The idea of generating more relevant samples as local explanations is interesting, but in my opinion this paper falls a bit short in demonstrating the value of relevance towards interpretability. The method appears to be a direct application of an existing neural net model, Semi-Quantized Activation Neural Network (SQANN), without significant modification. More clarity in the writing would also help this paper.

Nits:
- "damage control" term in the abstract --> a bit casual
- I found many of the figures too small and hard to read.
- Figure 4 presents some samples and asks the reader to compare to a particular figure in Koh and Liang 2017. This seems a bit unusual.

**Strength And Weaknesses:**

Strengths:
- The idea of generating more relevant samples as local explanations is interesting.

Weaknesses:
- No comparison to other post-hoc methods that generate samples as explanations, despite referencing them.
- Method performs worse than a CNN (table 1). This could be acceptable, since this is a post-hoc method. Standard deviations would help in judging the significance of these results.
- Interpretability evaluation a bit lacking -- are the sample explanations claimed to be more relevant generated by the method actually perceived as more relevant / helpful / interpretable by humans?

**Summary Of The Paper:**

The paper introduces a method called kaBEDONN that produces "relevant" samples as a local explanation.

**Summary Of The Review:**

The paper has an interesting idea of generating more relevant samples as local explanations. However, the empirical evaluation felt incomplete (see above), and a user study to link relevance to interpretability is missing.

---

### Official Review · Reviewer_gXuS · 2022-10-23

**Confidence:** 4
**Clarity, Quality, Novelty And Reproducibility:** See above.
**Correctness:** 1
**Technical Novelty And Significance:** 1
**Empirical Novelty And Significance:** 1
**Recommendation:** 1

**Strength And Weaknesses:**

**Strengths**

The paper tackles an important problem: the interpretability of neural networks.

The proposed architecture seems novel.

**Weaknesses**

The writing quality is poor. Generally, this does not feel like a well-polished academic paper. I would recommend the authors to ask others not involved in this project to read the paper and identify points of confusions.

It is not clear how this model supports the three objectives described in Sec. 1, especially the latter two, which seem to be not demonstrated.

There is no comparison with existing sample-importance methods in the experiment.

It is not clear what conclusions can be drawn from Fig. 4 results. It looks like main node often is the same as data sample. Why should this happen? If the data sample is from the test set, there shouldn't be a duplicate in the training set? It's not clear what additional insights we could get from the WR/Sub nodes either.

Why are the results in ImageNet not centrally highlighted? I would expect those to be much more interesting than that on MNIST and CIFAR, but most of the results are on the latter.

Minor:

I recommend the authors to familiarize themselves with latex commands `\citep` and `\citet`.

**Summary Of The Paper:**

This paper presents kaBEDONN, a method for post hoc explaining neural network classifiers by identifying relevant data. It is inspired by existing sample importance-based explanations, such as the influence function and representer point.

**Summary Of The Review:**

Due to the numerous issues with the current draft, I vote for rejection.

---

### Decision · Program_Chairs · 2023-01-20

**Decision:**

Reject

**Justification For Why Not Higher Score:**

Missing comparisons with related work.

**Justification For Why Not Lower Score:**

N/A

**Metareview: Summary, Strengths And Weaknesses:**

The authors introduce a system for postdoc querying of training data samples, which serve as explanations. The reviewers have pointed out serious issues with the writing, as well as a lack of comparisons with existing work. The authors responded that none of the existing methods are applicable here, but there should be some way of demonstrating this work is better than existing ones - a user study if nothing else, showing whether these explanations increase confidence in classification.
Overall, the paper is not publishable in its current form.